# A novel role for lipoxin $A_4$ in driving a lymph node–eye axis that controls autoimmunity to the neuroretina

**Jessica Wei[1,2], Mary J Mattapallil[2], Reiko Horai[2], Yingyos Jittayasothorn[2], Arnav P Modi[3], H Nida Sen[2], Karsten Gronert[1,3,4]\*, Rachel R Caspi[2]\***

[1]Vision Science Program, University of California, Berkeley, Berkeley, United States; [2]Laboratory of Immunology, National Eye Institute, National Institutes of Health, Bethesda, United States; [3]School of Optometry, University of California, Berkeley, Berkeley, United States; [4]Infectious Disease and Immunity Program, University of California, Berkeley, Berkeley, United States

**\*For correspondence:**
kgronert@berkeley.edu (KG);
caspir@nei.nih.gov (RRC)

**Competing interests:** The authors declare that no competing interests exist.

**Abstract** The eicosanoid lipoxin $A_4$ (LXA$_4$) has emerging roles in lymphocyte-driven diseases. We identified reduced LXA$_4$ levels in posterior segment uveitis patients and investigated the role of LXA$_4$ in the pathogenesis of experimental autoimmune uveitis (EAU). Immunization for EAU with a retinal self-antigen caused selective downregulation of LXA$_4$ in lymph nodes draining the site of immunization, while at the same time amplifying LXA$_4$ in the inflamed target tissue. T cell effector function, migration and glycolytic responses were amplified in LXA$_4$-deficient mice, which correlated with more severe pathology, whereas LXA$_4$ treatment attenuated disease. In vivo deletion or supplementation of LXA$_4$ identified modulation of CC-chemokine receptor 7 (CCR7) and sphingosine 1- phosphate receptor-1 (S1PR1) expression and glucose metabolism in CD4$^+$ T cells as potential mechanisms for LXA$_4$ regulation of T cell effector function and trafficking. Our results demonstrate the intrinsic lymph node LXA$_4$ pathway as a significant checkpoint in the development and severity of adaptive immunity.

## Introduction

Understanding the development of adaptive immune responses to self-antigens is imperative, as prevalence of autoimmunity has increased steadfastly in recent decades (*Lerner and Matthias, 2015*; *Bach, 2002*). Posterior autoimmune uveitis accounts for 10–15% of blindness in the Western world and primarily affects working age adults (*Durrani et al., 2004*; *Gritz and Wong, 2004*). Auto-immune uveitis is mediated by pathogenic Th1 and Th17 responses to retina-specific antigens, such as interphotoreceptor retinoid-binding protein (IRBP) and retinal arrestin. EAU in mice closely reca-pitulates human disease. Upon peripheral immunization with retinal antigens in complete Freund's adjuvant (CFA), T cells are primed in the inguinal lymph nodes that drain the immunization sites. Activated retina-specific T cells then traffic to the eye and elicit inflammation that starts 9–11 days and peaks 14–16 days after immunization (*Caspi, 2010*).

LXA$_4$ is an endogenous arachidonic acid-derived signaling molecule that exerts anti-inflammatory and pro-resolving activities via formyl peptide receptor 2 (ALX/FPR2) in mice and humans. To control the severity and duration of acute inflammation, cells expressing 5-lipoxygenase (5-LOX) and 12/15-lipoxygenase (12/15-LOX) oxygenate arachidonic acid in tandem to synthesize LXA$_4$ locally. 5-LOX is the only enzyme in mammals that can generate the 5,6 epoxide intermediate leukotriene $A_4$ required for the formation of LXA$_4$ and other epoxide intermediates of several specialized pro-resolving mediators (SPMs) (*Wei and Gronert, 2017*; *Serhan, 2014*). Even though 5-LOX also ini-tiates the generation of pro-inflammatory leukotrienes by leukocytes, 5-LOX in the retina has

protective roles (*Sapieha et al., 2011*; *Livne-Bar et al., 2017*). Emerging evidence indicates that LXA$_4$ is an important tissue-resident pathway in eyes and draining lymph nodes that controls ocular surface inflammation and immune responses, and it is neuroprotective in the retina (*Wei and Gronert, 2017*; *Livne-Bar et al., 2017*; *Kenchegowda and Bazan, 2010*).

The defining functions of LXA$_4$ include inhibition of neutrophil infiltration to the site of inflammation, promotion of macrophage efferocytosis, and reduction of inflammatory cytokine and chemokine production (*Serhan, 2014*; *Serhan et al., 2008*). In vitro studies demonstrated that LXA$_4$ and SPMs have a role in T follicular and helper cell differentiation (*Nagaya et al., 2017*; *Chiurchiù et al., 2016*), and neutrophil-derived LXA$_4$ was previously shown to inhibit T effector cell responses in lymph nodes in immune-driven dry eye disease (*Gao et al., 2015*). ALX/FPR2 is expressed on T cells (*Ariel et al., 2003*), providing a modality for LXA$_4$ regulation of adaptive immunity. However, role of LXA$_4$ and its direct action on T lymphocytes in autoimmune responses has been minimally explored.

Specific aim of this study was to uncover whether and how LXA$_4$ regulates T effector cell function in the pathogenesis of autoimmune uveitis. Here, we report that expression of the homeostatic LXA$_4$ pathway and endogenous LXA$_4$ formation are dynamic and temporally defined in the target tissue, that is the eye, and the inciting lymph nodes that drain the site of immunization. T cells from LXA$_4$-deficient (*Alox5$^{-/-}$*) mice exhibited enhanced effector function with augmented cytokine production and altered metabolism that resulted in exacerbated disease, while LXA$_4$ treatment in wild type (WT) mice attenuated development of autoimmune uveitis. The resident LXA$_4$ pathway in draining lymph nodes was identified as a significant regulator of CCR7 and S1PR1, which control egress of T effector cells from lymph nodes. Lastly, LXA$_4$ deficiency as well as LXA$_4$ treatment regulated T cell glycolysis, an essential metabolic pathway required for T effector cell function. Our findings identify novel mechanisms and a homeostatic role for LXA$_4$ in lymph nodes to control the development of adaptive autoimmune responses.

## Results

### LXA$_4$ is generated during autoimmune uveitis in a time- and site-dependent manner

Changes of SPMs in human blood have been proposed as diagnostic markers of health and disease outcomes (*Colas et al., 2014*; *Norris et al., 2018*). Postmortem uveitic retinas are not readily available considering uveitis is not a fatal condition, therefore, we analyzed serum samples from 41 healthy subjects and 78 uveitis patients using ELISA to assess whether LXA$_4$ levels are affected in patients diagnosed with posterior uveitis (*Figure 1A*). The 78 uveitis patients had posterior uveitis that was either standalone or associated with various primary conditions such as Vogt-Koyanagi-Harada disease (VKH), birdshot chorioretinopathy, panuveitis, intermediate uveitis, choroidopathy and retinopathy. Average serum level of LXA$_4$ in patients were significantly lower than in healthy subjects (p=0.0135). Most uveitis diagnoses (n = 2 to 23 per diagnosis) had mean serum LXA$_4$ levels that were lower than healthy controls, with the exception of VKH. However, differences between individual diagnoses and healthy subjects did not reach significance, except for the group of intermediate uveitis (n = 5, p=0.019). The overall decrease in serum LXA$_4$ level in patients suggests that the LXA$_4$ pathway is physiologically relevant during pathogenesis of posterior segment uveitis.

To investigate the role of LXA$_4$ in posterior autoimmune uveitis, we induced EAU in C57BL/6J WT mice (*Caspi, 2010*; *Caspi, 2003*) and quantified LXA$_4$ and pathway-specific metabolite levels in the eye, submandibular lymph nodes, distal lymph nodes and inguinal lymph nodes that drain the immunization sites. Samples were collected from naive and immunized mice at disease onset (day 10) and peak disease (day 16) (*Figure 1B and C*). LXA$_4$ and its 5-LOX and 12/15-LOX pathway markers (5-HETE and 15-HETE) were significantly elevated in eyes at peak disease compared to naive unimmunized mice (*Figure 1C*). By contrast, LXA$_4$, 5-HETE and 15-HETE levels were significantly downregulated at peak disease in the inguinal lymph nodes (*Figure 1B and C*). LXA$_4$ levels did not change in the distal lymph nodes or eye-draining submandibular lymph nodes. Serum was analyzed at onset and peak of EAU (*Figure 1—figure supplement 1A*) to determine if the induced autoimmune response in mice would replicate changes in serum LXA$_4$ observed in uveitis patients (*Figure 1A*). While serum LXA$_4$ levels in EAU-challenged mice did not change compared to naïve mice, pathway markers 5-LOX and 15-LOX showed significant and progressive decreases during EAU (naïve vs.

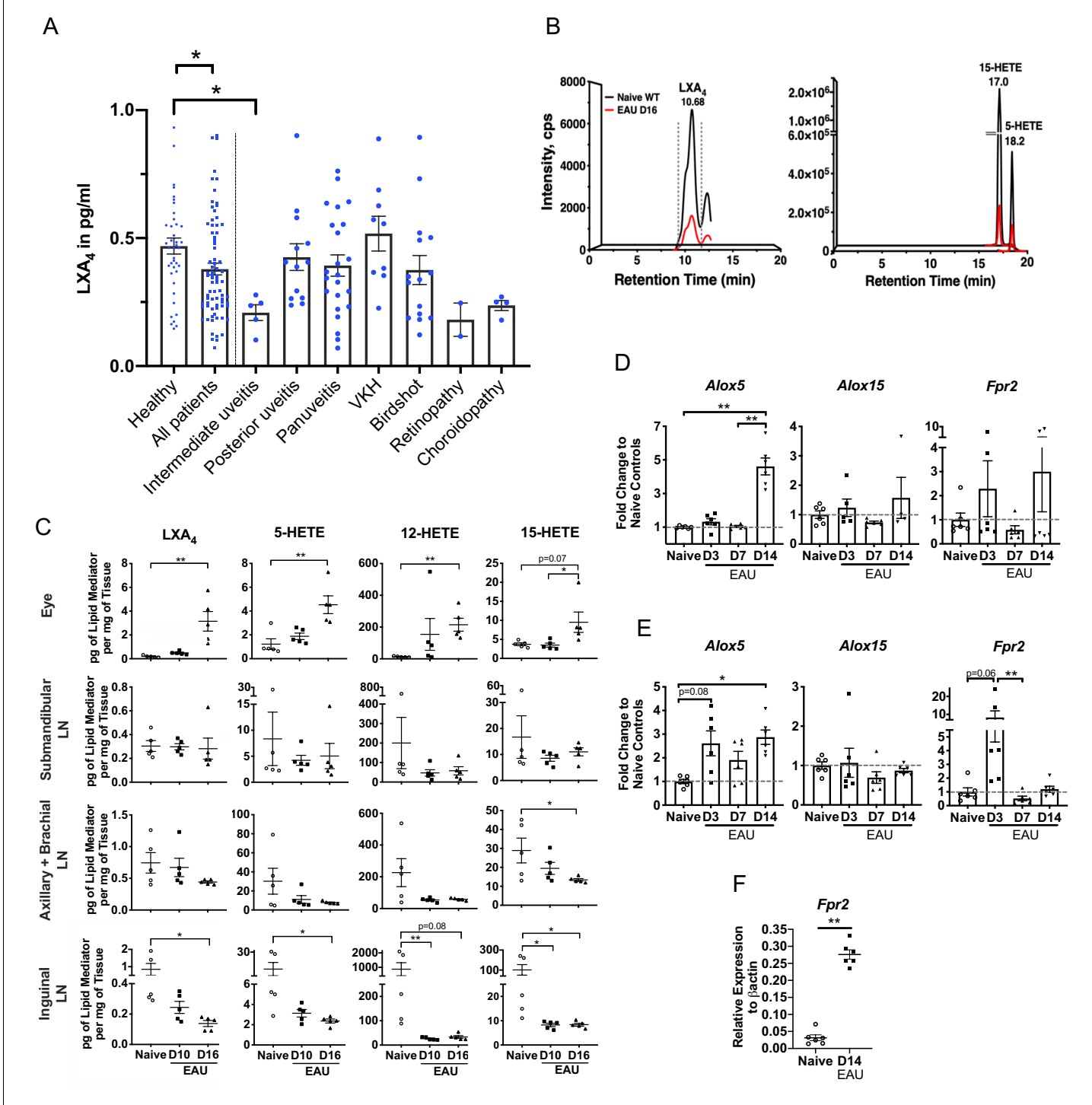

**Figure 1.** LXA$_4$ is generated during autoimmune uveitis in a disease- and tissue- dependent manner. (**A**) Bar graph of LXA$_4$ concentration in pg/ml measured in human serum by ELISA. Healthy subjects, n = 41; all uveitis patients, n = 78; and breakdown of uveitis diagnoses within all patient samples, n = 2 to 23 per diagnosis. C57BL/6J mice were immunized with 300 μg IRBP$_{651-670}$ and tissues collected for analyses on days 10 and 16 post-immunization for panels B-I. (**B**) Lipidomic analysis of inguinal lymph nodes from naïve and EAU-challenged mice on day 16. Peaks are representative multiple reaction monitoring LC-MS/MS chromatograms for specific ion transitions: LXA$_4$ (*m/z* 351 > 115), 5-HETE (*m/z* 319 > 115), 15-HETE (*m/z* 319 > 175). (**C**) LXA$_4$ and its pathway markers in pg per mg of tissue in whole eye globes, submandibular lymph nodes, distal (axillary + brachial) lymph nodes, and inguinal lymph nodes quantified by LC-MS/MS from unimmunized naïve and EAU-challenged mice (days 10 and 16). n = 5 per group. (**D–E**) Temporal expression of *Alox5, Alox15* and *Fpr2* in (**D**) retinas, and (**E**) inguinal lymph nodes during EAU (days 3, 7, 14) in comparison to the respective

*Figure 1 continued on next page*

Figure 1 continued

tissue from naïve mice quantified by RT-PCR. n = 6 per group. (**F**) *Fpr2* expression on CD4$^+$ T cells isolated from inguinal lymph nodes of naive and immunized mice, n = 6 per group. *p<0.05, **p<0.01, One-way ANOVA and Mann-Whitney test.

The online version of this article includes the following figure supplement(s) for figure 1:

**Figure supplement 1.** Murine serum LXA$_4$ level and in vivo LTB$_4$ formation during EAU pathogenesis.

EAU day 16, p=0.0078 and p=0.0048 for 5-HETE and 15-HETE respectively). Analytes in lipidomic analysis also included DHA- and EPA-derived SPMs and leukotrienes. Pathway markers for DHA-derived SPMs (4-HDHA, 7-HDHA, 14-HDHA and 17-HDHA) were detected in all tissues, but DHA- or EPA- derived SPMs were not robustly detected or were below the signal-to-noise threshold (5:1) in our method. Leukotriene B$_4$ (LTB$_4$), a 5-LOX product, was detected in lymph nodes of healthy mice and during the time course of EAU (*Figure 1—figure supplement 1B*). However, unlike LXA$_4$, LTB$_4$ levels did not change significantly in inguinal lymph nodes during EAU pathogenesis. The finding is consistent with our previous lipidomic analysis that identified changes in LXA$_4$, but not LTB$_4$, in eye draining lymph nodes of an immune-driven dry eye disease model (*Gao et al., 2015*; *Gao et al., 2018*). Altogether, the current findings indicate selective and differential regulation of LXA$_4$ formation at inductive and effector sites of autoimmunity in EAU.

We next assessed gene expression of the LXA$_4$ pathway during EAU. Retinas and inguinal lymph nodes were harvested from naïve and immunized mice on day 3, day 7, and day 14 post-immunization. Expression of 5-LOX (*Alox5*), 12/15-LOX (*Alox15*), and LXA$_4$ receptor (*Fpr2*) was measured by RT-PCR. In the retina, *Alox5* expression was upregulated by approximately five-fold on day 14 post-immunization in comparison to earlier time points and to naïve mice (*Figure 1D*), which directly correlated with upregulation of LXA$_4$ formation in the eye. *Alox15* expression remained comparable to naïve mice during the entire time course of EAU, whereas *Fpr2* expression changed throughout the time course of EAU, likely reflecting changes in circulating leukocytes populations that express varying levels of *Fpr2* in the lymph node, and in temporal regulation of the homeostatic LXA$_4$ pathway by resident cells in the retina.

In the inguinal lymph nodes, *Alox5* expression was significantly upregulated on day 3 and day 14 post-immunization, whereas *Fpr2* expression was highly upregulated on day 3, but not on day 14 post-immunization (*Figure 1E*). Increased RNA levels of *Alox5* in inguinal lymph nodes did not correlate with 5-LOX activity and LXA$_4$ production as both 5-HETE and LXA$_4$ levels were decreased on day 10 and day 16 (*Figure 1C*). Since EAU pathogenesis is mediated by retina-specific CD4$^+$ T cells, we evaluated *Fpr2* expression on CD4$^+$ T cells isolated from inguinal lymph nodes and found that the receptor expression was significantly upregulated at peak disease in comparison to CD4$^+$ T cells isolated from inguinal lymph nodes of naïve mice (*Figure 1F*). RNA expression of lipoxygenases (5-LOX and 12/15-LOX) demonstrates selective and temporal modulation of tissue-resident LXA$_4$ pathway during EAU, while increased *Fpr2* expression in CD4$^+$ T cells in the inguinal lymph nodes potentiates their ability to interact with LXA$_4$ during the effector phase of the immune response.

## Endogenous LXA$_4$ limits development and progression of EAU

In light of LXA$_4$ pathway modulation observed during EAU, we investigated whether endogenous LXA$_4$ deficiency would affect the course of autoimmune uveitis. *Alox5*$^{-/-}$ mice are LXA$_4$ deficient (*Aliberti et al., 2002*), since 5-LOX is a necessary enzyme required for LXA$_4$ biosynthesis (*Figure 2A*). The *Alox5*$^{-/-}$ mouse line is an established Jackson Laboratory strain that has been used to study the role of leukotrienes and LXA$_4$ in vivo. There are no reports of compensatory upregulation of other eicosanoid pathways (cyclooxygenases, 12-LOX or 15-LOX) in this mouse line. Lipidomic analysis did not detect significant differences in basal levels of 12-HETE, 15-HETE, prostaglandin D$_2$ (PGD$_2$), prostaglandin E$_2$ (PGE$_2$), or thromboxane B$_2$ (TXB$_2$) in inguinal lymph nodes of *Alox5*$^{-/-}$ mice (*Figure 2—figure supplement 1*). Low amounts of 5-HETE and LXA$_4$ were detected in *Alox5*$^{-/-}$ mice (*Figure 2A*). It is possible that one of the other mouse lipoxygenase enzymes, which do not exist in humans (e.g. 8-LOX), is able to catalyze LXA$_4$ formation in mice with low efficiency. Age-matched *Alox5*$^{-/-}$ and WT mice were immunized with IRBP$_{651-670}$ peptide (*Mattapallil et al., 2015*) and EAU pathogenesis was assessed by fundoscopy. Disease scores of *Alox5*$^{-/-}$ mice were significantly higher than WT mice starting on day 16 post-immunization, and retinal inflammation in

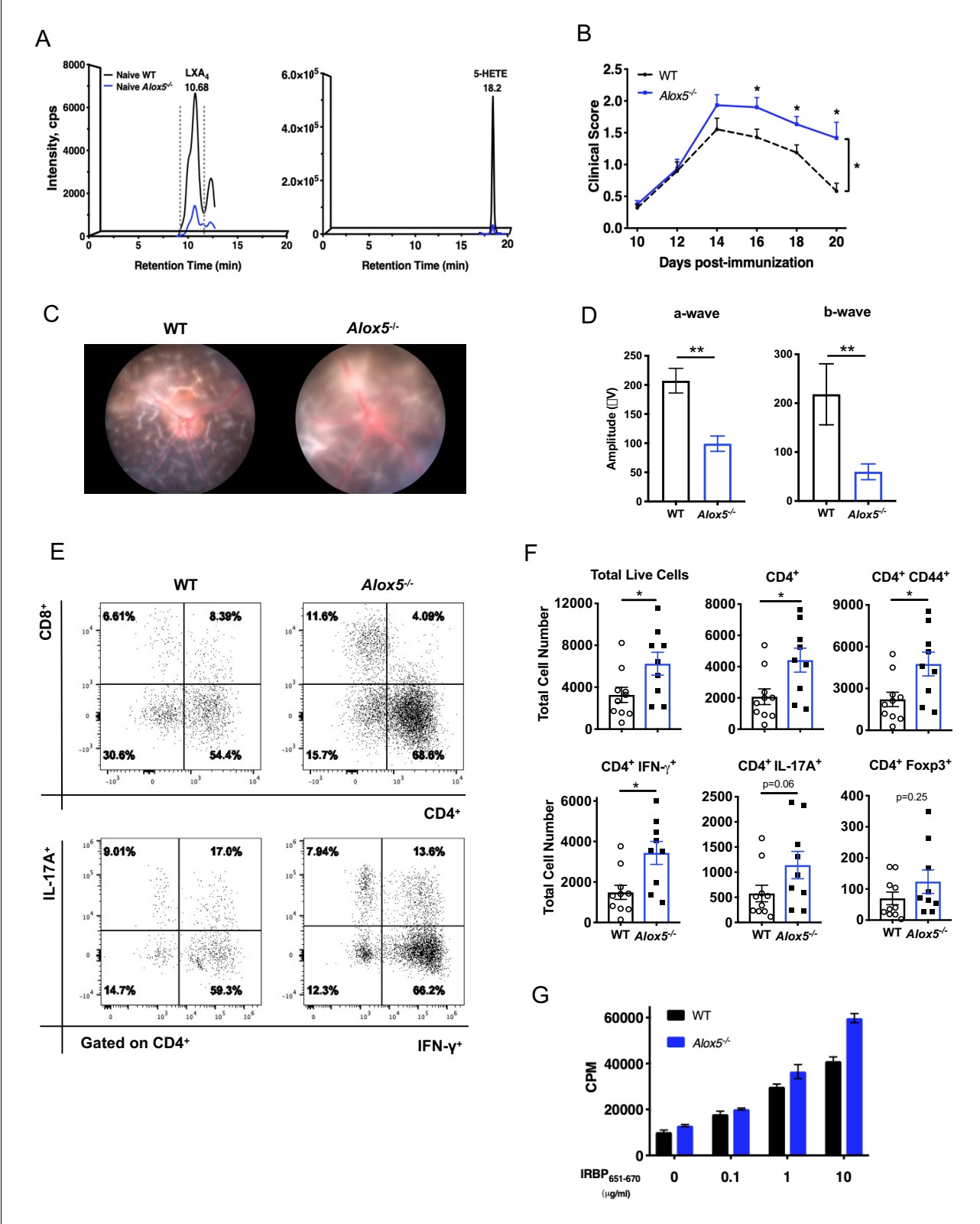

**Figure 2.** LXA$_4$ deficiency exacerbates disease progression. (**A**) Chromatograms from LC-MS/MS analyses showing LXA$_4$ and 5-HETE formation in naïve WT and *Alox5*$^{-/-}$ mice. (**B**) Fundoscopy scores of WT and *Alox5*$^{-/-}$ mice immunized with 150 µg of IRBP$_{651-670}$. n = 15 per group through day 18, n = 6–9 per group on day 20. Combined from two experiments. (**C**) Representative Micron III fundus images of immunized WT and *Alox5*$^{-/-}$ mice taken on day 14 post-immunization. (**D**) Electroretinography of EAU-challenged WT and *Alox5*$^{-/-}$ mice measured on day 13 post-immunization following dark

*Figure 2 continued on next page*

Figure 2 continued

adaptation, n = 10 eyes per group. (E) Representative flow plots of CD4$^+$ vs. CD8$^+$ and IFN-γ vs. IL-17A producing CD4$^+$ cells in the eyes of WT and *Alox5$^{-/-}$* mice. (F) Flow cytometry analyses of total numbers of ocular infiltrates in EAU-challenged WT and *Alox5$^{-/-}$* gated on live cells, CD4$^+$, CD4$^+$ CD44$^+$, CD4$^+$ IFN-γ$^+$, CD4 IL-17A$^+$ populations, n = 9–10 mice per group. (G) Representative bar graphs of antigen-specific cell proliferation performed on inguinal lymph nodes harvested on day 18 post-immunization from WT and *Alox5$^{-/-}$* mice, pooled n = 6–9 mice per group. Showing one representative from three experiments with the same trend. *p<0.05, **p<0.01, Wilcoxon matched-pairs signed rank test and unpaired t test.
The online version of this article includes the following figure supplement(s) for figure 2:

**Figure supplement 1.** In vivo generation of other eicosanoids in WT and *Alox5$^{-/-}$* mice.

*Alox5$^{-/-}$* persisted while inflammation gradually resolved in WT by day 20 post-immunization (*Figure 2B*). Representative Micron III fundus images of WT and *Alox5$^{-/-}$* mice revealed more inflammation in the *Alox5$^{-/-}$* eye, with retinal haze resulting from severe cellular infiltration in the retina and vitreous, as opposed to clear vitreous in the WT eye allowing visualization of the underlying retinal folds (*Figure 2C*). Electroretinography showed that visual function of immunized *Alox5$^{-/-}$* mice was significantly impaired compared to WT mice (*Figure 2D*), in line with the more severe disease phenotype in *Alox5$^{-/-}$* mice. Flow cytometric analyses of ocular infiltrating cells revealed that there were significantly more cell infiltrates including CD4$^+$ T cells and CD4$^+$ IFN-γ producing T cells in eyes of *Alox5$^{-/-}$* mice than in WT mice on day 18 post-immunization (*Figure 2E,F*). Lymphocytes from *Alox5$^{-/-}$* mice exhibited higher antigen-specific proliferative responses to IRBP$_{651-670}$ peptide than WT mice, as assessed by $^3$H thymidine-incorporation assay (*Figure 2G*). These results demonstrate more severe EAU development in the target tissue and at the site of T cell priming in the absence of LXA$_4$.

## LXA$_4$ treatment attenuates development of uveitis

We next assessed if amplifying the LXA$_4$ pathway could limit EAU progression. To this end, WT mice were immunized with IRBP$_{651-670}$ peptide and were treated with either 1 μg of LXA$_4$ or vehicle control daily, starting on the day of immunization. LXA$_4$-treated immunized mice developed significantly less disease compared to the vehicle-treated group (*Figure 3A*). Retinal histology assessed by H and E stained sections confirmed that LXA$_4$ treatment conferred protection in comparison to vehicle-treated group (*Figure 3B and C*). The retinas of vehicle-treated group exhibited more severe cellular infiltration that contributed to retinal folds. In contrast, the posterior segment of eyes from LXA$_4$-treated mice maintained largely normal retinal morphology with no inflammatory exudate resembling that of naïve unimmunized mice, despite presence of intravitreal cell infiltrates consistent with a mild form of uveitis (*Figure 3C*). Inducible cyclooxygenases-2 (COX-2), a hallmark of many inflammatory diseases, was used as an inflammation marker for the retina. Immunohistochemistry revealed COX-2 localization in the outer plexiform, choroid, retinal pigmented epithelium (RPE) and photoreceptor layers of the eye, while CD4$^+$ T cell infiltration was observed in the choroid, RPE and into the photoreceptor layers (*Figure 3D*). In LXA$_4$-treated mice, the number of CD4$^+$ T cells and the level of COX-2 expression were significantly reduced in comparison to vehicle-treated mice (*Figure 3E*).

Th1 and Th17 are T cell subsets that drive EAU pathogenesis (*Damsker et al., 2010*). Flow cytometry analyses of eyes harvested during peak inflammation confirmed that LXA$_4$ treatment significantly reduced ocular infiltration of pathogenic IFN-γ and IL-17A producing CD4$^+$ T cells (*Figure 3F and G*). Immune cells from inguinal lymph nodes of vehicle-treated mice proliferated dose-dependently after IRBP$_{651-670}$ peptide challenge. In contrast, antigen-specific recall responses of lymphocytes from inguinal lymph nodes of LXA$_4$-treated mice were markedly decreased (*Figure 3H*).

Since LXA$_4$ treatment reduced antigen-specific recall responses, we investigated whether LXA$_4$ could regulate disease during the effector phase. We used in vitro-activated T cells from T cell receptor transgenic mice that express a receptor specific for IRBP$_{161-180}$ (R161H) on the B10. RIII background and develop spontaneous uveitis (*Horai et al., 2015*). Lymph node T cells from CD90.1 congenic R161H mice were cultured with IRBP$_{161-180}$ peptide for three days in vitro, which resulted in ~70% of IFN-γ producing T cells (data not shown). IRBP-primed cells were injected into B10.RIII WT CD90.2 recipient mice and received either LXA$_4$ or vehicle control treatment starting on the day of cell transfer. LXA$_4$-treated mice developed significantly lower disease in a dose-dependent manner than vehicle-treated mice (*Figure 3I*). These findings provide direct evidence that LXA$_4$ inhibits

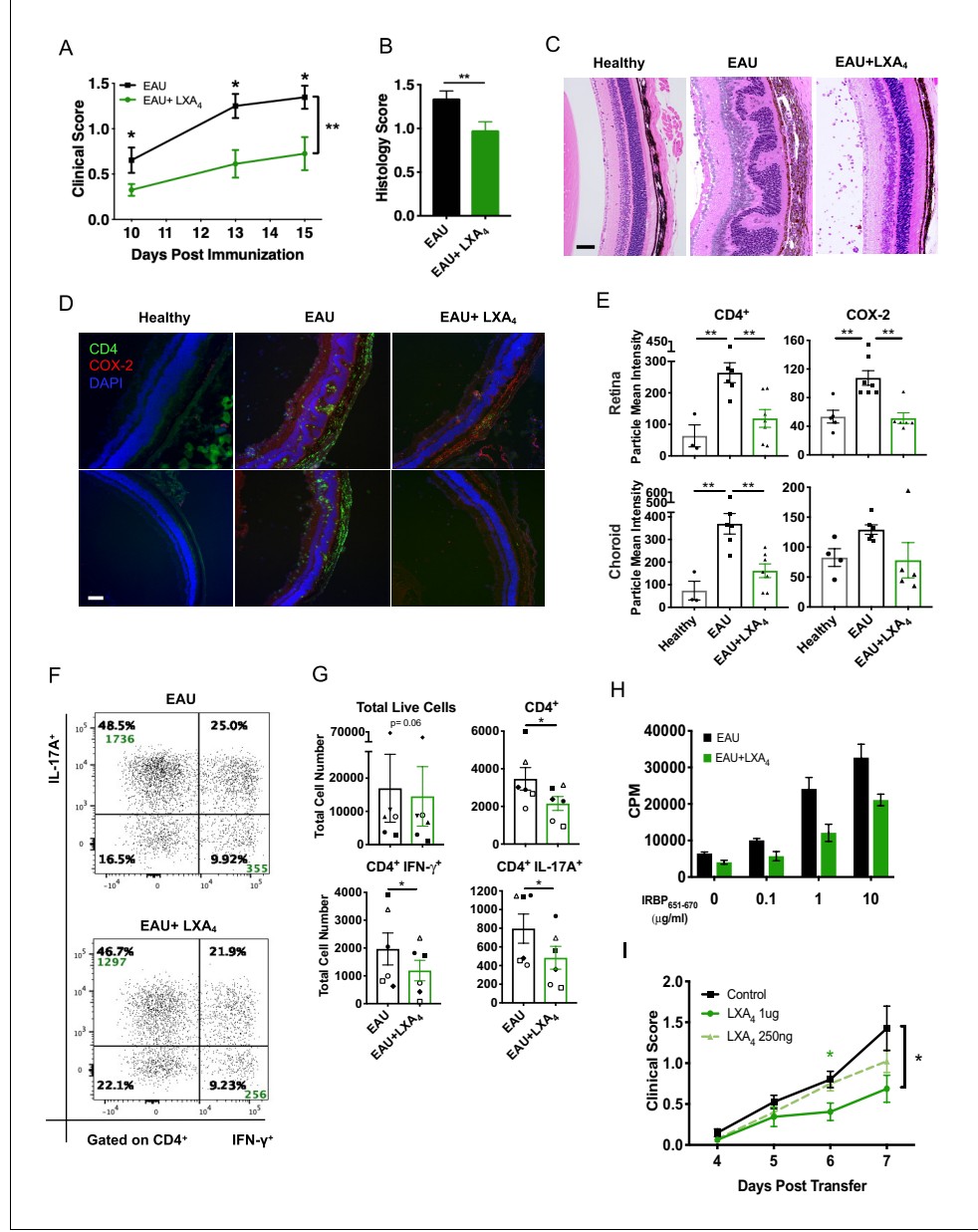

**Figure 3.** LXA$_4$ treatment attenuates the development of uveitis. Mice were immunized with 300 μg of IRBP$_{651-670}$ to induce EAU and treated with LXA$_4$ or vehicle control starting on the day of immunization. (**A**) Clinical scores of retinal inflammation assessed by fundoscopy, n = 10 per group. (**B**) Histology scores of eyes at peak inflammation, n = 24 per group combined from three experiments. (**C**) Representative H and E staining of the posterior segment of the eye. 20X, scale bar = 50 μm. (**D**) Representative immunohistochemistry images of eyes harvested at peak inflammation. Top row: central retina, bottom row: peripheral retina. 10X, scale bar = 100 μm. (**E**) Fiji quantification of CD4 and COX-2 fluorescence particle intensity of immunohistochemistry images, n = 3–7 per group. (**F**) Representative flow plots of eyes harvested at peak inflammation. Numbers in green are the total cell counts in the gated population. (**G**) Total number of ocular infiltrates at peak inflammation, n = 6 experiments (pooled 4–6 mice per group). (**H**) Bar graphs of proliferation assay performed on lymph nodes harvested at peak inflammation, pooled n = 6 mice per group. Representative assay of 3 separate experiments. (**I**) Fundoscopy scores of a representative adoptive transfer experiment out of 3. 5 × 10$^6$ R161H CD90.1 lymphocytes were activated with antigen IRBP$_{161-180}$ in vitro for 3 days and transferred into WT CD90.2 recipients. Recipients were treated with indicated doses of LXA$_4$ starting on the day of cell transfer. *p<0.05, **p<0.01, Mann-Whitney test and One-way ANOVA.

the actions of antigen-primed lymphocytes in vivo and also regulates the effector phase of the adaptive immune response.

## LXA$_4$ modulates T cell trafficking in inguinal lymph nodes that drain the immunization site

To identify the genes that are regulated by LXA$_4$ treatment, we compared RNA expression in eyes and peripheral lymph nodes from immunized mice treated with LXA$_4$ or vehicle. Tissues were harvested on day 16 post-immunization and gene expression analyzed by nanoString. Eyes of vehicle-treated mice exhibited increased expression of inflammatory markers *Icam1, Il1b, Mapk1, Mapk3, Nfkb1, Ptgs2* (*Cox-2*), *Tgfb, Tnfaip3,* and *Vcam1* in eyes, whereas LXA$_4$ treatment downregulated the expression of these markers during EAU. By contrast, expression of these genes was comparable in the inguinal lymph nodes of vehicle *vs.* LXA$_4$-treated animals (**Figure 4A**). These data suggest that systemic LXA$_4$ treatment reduced expression of inflammatory genes in the effector site, that is eyes, but not in the inductive site, that is lymph nodes draining the site of immunization.

Because LXA$_4$ treatment reduced T cell migration into the eye (**Figure 3G**), we examined gene expression of the T cell trafficking marker *Ccr7* and its cognate ligands *Ccl19* and *Ccl21a*, as well as the Th1-associated chemokine receptor *Cxcr3*, and Th17-associated chemokine receptor *Ccr6*. Compared to vehicle- treated mice, eyes of LXA$_4$ treated mice had downregulated expression of *Ccr7, Cxcr3* and *Ccl19* (**Figure 4B**), which paralleled the reduced numbers of total CD4$^+$ T cells and CD4$^+$ IFN-$\gamma^+$ T cells in the ocular infiltrate (**Figure 3G**). In contrast, *Ccr7* and *Ccl21a* were highly upregulated in the inguinal lymph nodes of LXA$_4$-treated mice compared to controls (**Figure 4B**). We next assessed whether LXA$_4$ treatment also impacts expression of *S1pr1*, which regulates lymphocyte egress from lymphoid tissues (**Garris et al., 2014**). LXA$_4$ treatment downregulated expression of *S1pr1* in the inguinal lymph nodes that drain the site of immunization, but not in the eye, submandibular lymph nodes that drain the eye, or distal lymph nodes (**Figure 4C**). To examine whether downregulation of *S1pr1* under conditions of LXA$_4$ deficiency can be detected in uveitis-relevant T cells, we assessed *S1pr1* expression in CD4$^+$ T cells from inguinal lymph nodes of EAU-challenged *Alox5$^{-/-}$* and WT mice by RT-PCR. *S1pr1* expression showed a trend of higher expression in CD4$^+$ T cells from *Alox5$^{-/-}$* mice (**Figure 4D**). The opposing expression of lymphocyte migration-associated receptors and ligands between the eyes and the inguinal lymph nodes identifies a potential mechanism for LXA$_4$ to control the magnitude of an adaptive immune response, specifically by regulating of *Ccr7* and *S1pr1* expression that mediate T cell egress from lymph nodes.

Next, we assessed whether LXA$_4$ deficiency impacts migration of CD4$^+$ T cells from secondary lymphoid tissues in vitro. CD4$^+$ T cells were isolated from inguinal lymph nodes and spleens of EAU-challenged WT and *Alox5$^{-/-}$* mice on day 13 post-immunization and stimulated in vitro with anti-CD3 and anti-CD28 for 18 hr prior to being subjected to a transwell migration assay. Significantly higher numbers of CD4$^+$ T cells from LXA$_4$-deficient mice migrated toward CCL19 and CCL21 chemokine gradients than CD4$^+$ T cells from WT mice (**Figure 4E**). These results suggest that the absence of tissue-resident LXA$_4$ increases migration of activated lymph node T cells and may explain exacerbated EAU in LXA$_4$-deficient mice (**Figure 2**). To confirm the direct regulation of chemokine receptors on T cells by LXA$_4$, S1PR1 and CCR7 expression was assessed by flow cytometry on in vitro stimulated T cells treated with LXA$_4$. CCR7 expression was modestly upregulated (**Figure 4F**) while S1PR1 expression was downregulated (**Figure 4G**) in a dose-dependent manner. Together, these data are compatible with the interpretation that LXA$_4$ acts directly on T cells and modulates trafficking through regulation of S1PR1 and CCR7 expression.

## Disease-limiting effect of LXA$_4$ is T cell-mediated

Next, we examined whether enhanced EAU in the absence of LXA$_4$ is a T cell-mediated effect. We immunized WT and *Alox5$^{-/-}$* donor mice to expand IRBP-specific T cells, and on day 11 purified CD3$^+$ T cells from inguinal lymph nodes and spleens. The donor T cells were transferred into *TCRb$^{-/-}$* mice, which lack their own T cells, and three weeks after transfer the recipients were challenged for EAU with IRBP$_{651-670}$ peptide (**Figure 5A**). Consistent with the phenotype of exacerbated EAU in *Alox5$^{-/-}$* mice (**Figure 2**), recipients of *Alox5$^{-/-}$* T cells developed more severe EAU, with more granulomatous lesions and damage in the photoreceptor cell layer, than recipients of WT T cells

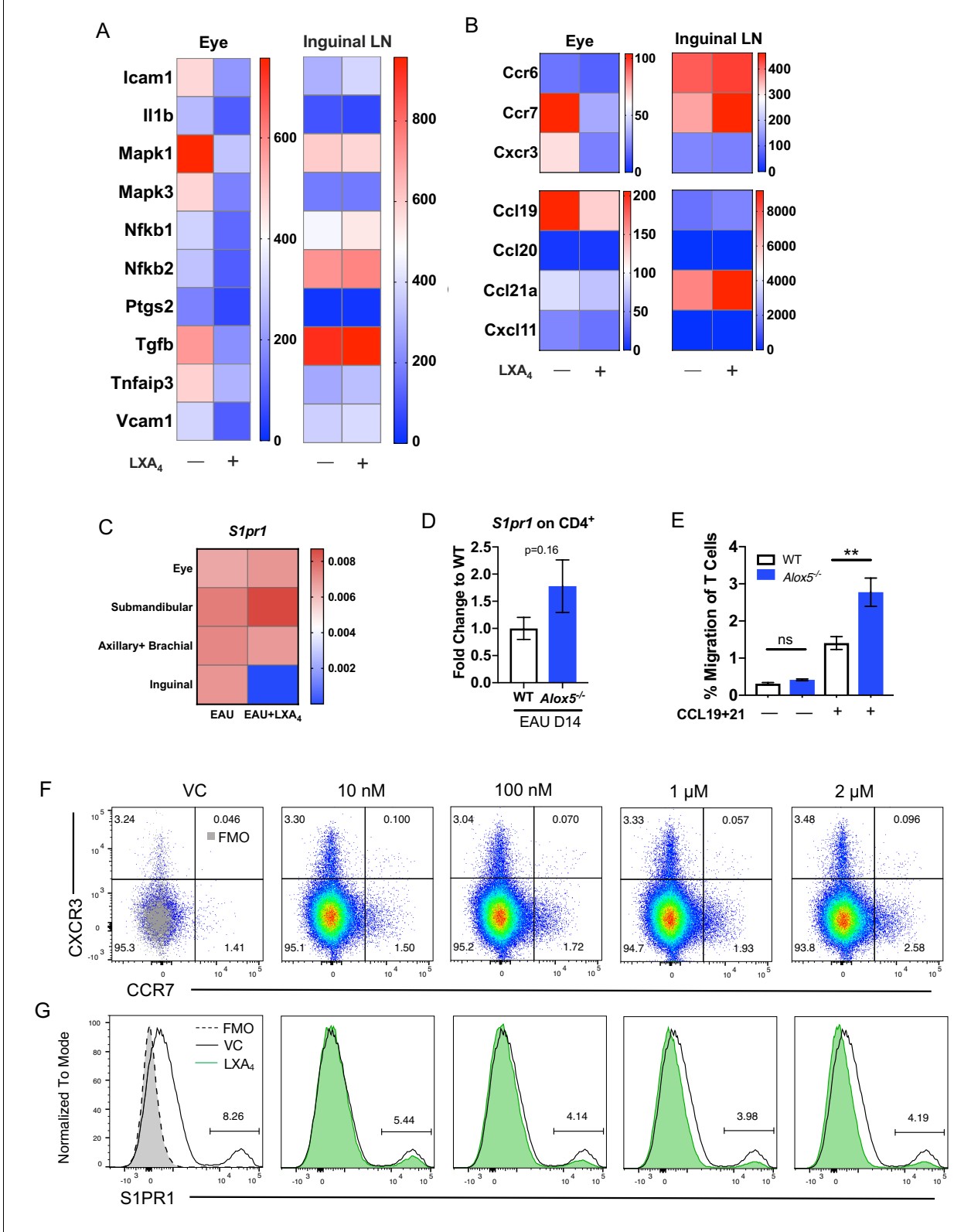

**Figure 4.** LXA$_4$ modulates T cell trafficking in inguinal lymph nodes that drain the immunization site. All tissues were collected from mice immunized with 150 μg of IRBP$_{651-670}$. (**A–B**) Heat maps showing gene expression of inflammation and cell trafficking markers in the eyes and inguinal lymph nodes harvested from EAU-challenged mice treated with vehicle or LXA$_4$. (**A**) Inflammation markers and (**B**) migration markers analyzed by nanoString. Each sample was pooled from 3 to 4 mice on day 16 post-immunization. (**C**) *S1pr1* expression relative to *β-actin* from eyes and various lymph nodes of mice

*Figure 4 continued on next page*

*Figure 4 continued*

treated with vehicle or LXA$_4$ harvested on day 16 post-immunization, same samples as panels A and B. (D) Fold change in *S1pr1* expression of CD4$^+$ T cells isolated from WT and *Alox5$^{-/-}$* on day 14 post-immunization, n = 11 from three experiments combined. (E) Transwell migration assay of CD4$^+$ T cells isolated from immunized WT and *Alox5$^{-/-}$* on day 13 post-immunization. CD4$^+$ T cells were pre-stimulated with anti-CD3 and anti-CD28 antibodies for 18 hr and transferred to transwell culture plates and incubated for 4 hr in the absence or presence of CCL19 and CCL21, n = 12 per group. One representative experiment of 4 showing the same trend. **p<0.01, unpaired Welch's t test. (F–G) Representative flow plots of CCR7 and S1PR1 expression on in vitro anti-CD3 and anti-CD28 stimulated CD4$^+$ T cells treated with vehicle control or 10 nM, 100 nM, 1 µM, and 2 µM of LXA$_4$. Cells were gated on live single cells then CD4$^+$ cells. Gray overlays indicate fluorescence minus one controls. Showing one representative experiment out of three.

(*Figure 5B and C*). Similar results, more severe EAU, were also observed in *TCRb$^{-/-}$* recipients of Fpr2$^{-/-}$ T cells (*Figure 5—figure supplement 1*).

To elucidate effector T cell phenotype and expression of trafficking molecules on uveitogenic CD4$^+$ donor T cells, transferred T cells were harvested from eyes and inguinal lymph nodes of *TCRb$^{-/-}$* recipient mice on day 20 post-immunization and analyzed by flow cytometry. Eyes of *Alox5$^{-/-}$* T cell recipients had significantly more total ocular infiltrates, specifically total CD4$^+$ T cells and IFN-γ producing CD4$^+$ T cells than WT donor T cell recipients (*Figure 5D*). However, there was no apparent difference in expression of the trafficking markers CXCR3, CCR6 and CCR7 by T cells that had infiltrated the eye (*Figure 5E*). In the inguinal lymph nodes, the frequency of IFN-γ$^+$ donor CD4$^+$ T cells from *Alox5$^{-/-}$* mice, but not total or IL-17A$^+$ donor CD4$^+$ T cells, was higher than that from WT (*Figure 5F*). Migration marker analysis showed higher frequency of CXCR3$^+$ in *Alox5$^{-/-}$* donor CD4$^+$ T cells in the inguinal lymph nodes, whereas frequency of activated CD4$^+$ CCR7$^+$ donor T cells from *Alox5$^{-/-}$* mice was significantly lower (*Figure 5G*). The downregulation of CCR7 expression and increased ocular infiltration of CD4$^+$ T cells support the hypothesis that egress of effector T cells from inguinal draining lymph nodes is augmented in LXA$_4$-deficient mice.

## LXA$_4$ pathway regulates the metabolism of CD4$^+$ T cells

T cell metabolic programming closely shapes T cell development and responses. Upon TCR stimulation, activated T cells engage in aerobic glycolysis to support effector T cell function (*Almeida et al., 2016*; *Buck et al., 2015*). We investigated whether LXA$_4$ deficiency impacts effector T cell metabolism since our findings indicate that the lymph node resident LXA$_4$ pathway regulates effector T cell function and migration (*Figure 4E* and *5G*). CD4$^+$ T cells were isolated from inguinal lymph nodes and spleens of EAU-challenged WT and *Alox5$^{-/-}$* mice and were stimulated for 18 hr with anti-CD3 and anti-CD28 antibodies. Cell metabolism was analyzed by the Seahorse Glycolytic Rate Assay. To accurately quantify glycolysis-derived extracellular acidification rate, mitochondrial $CO_2$-contributed acidification was measured and subtracted from total proton efflux rate, yielding the glycolytic proton efflux rate (glycoPER). CD4$^+$ T cells from *Alox5$^{-/-}$* mice achieved significantly higher glycoPER at baseline as well as after inhibition of the mitochondrial electron transport chain with rotenone and antimycin A (Rot/AA) (*Figure 6A–C*). Seahorse Glycolytic Rate assay was also used to assess the direct effect of LXA$_4$ on T cell glycolysis. CD4$^+$ T cells from WT mice were treated in vitro with LXA$_4$ or vehicle control. LXA$_4$ treated CD4$^+$ T cells had significantly lower glycoPER at base line and after mitochondrial inhibition (*Figure 6D*), as well as lower basal (*Figure 6E*) and compensatory (*Figure 6F*) glycolysis. Thus, both ex vivo evidence from CD4$^+$ T cells of EAU-challenged LXA$_4$-deficient mice and in vitro experiments of LXA$_4$-treated CD4$^+$ WT T cells indicate that lymph node resident LXA$_4$ has a direct role in regulating metabolic functions of CD4$^+$ T cells.

## Discussion

In this study, we demonstrate that the resident LXA$_4$ pathway in lymph nodes is important in regulating T cell effector functions during development and progression of ocular autoimmunity. A key finding is the downregulation of basal LXA$_4$ levels in lymph nodes that accompany the adaptive immune response in a mouse model of autoimmune uveitis. Reduced levels of serum LXA$_4$ were also identified in serum of human patients with posterior uveitis diagnoses, suggesting that the LXA$_4$ pathway is dynamically regulated in human uveitis. In contrast, LXA$_4$ levels were elevated in the uveitic eyes of mice, which is consistent with the established role of LXA$_4$ as an induced counter-regulatory

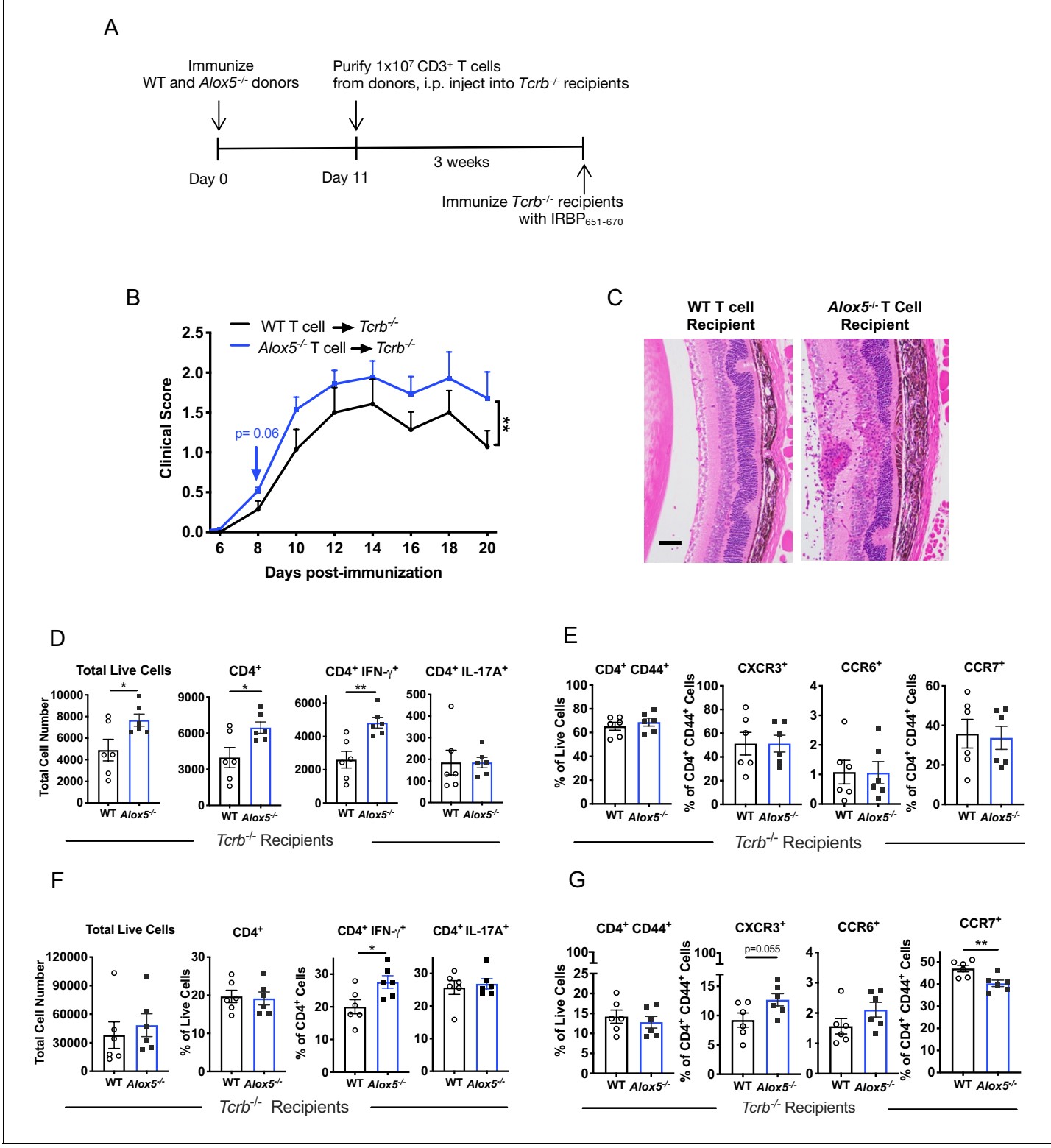

**Figure 5.** Disease limiting effect of LXA₄ is T cell-mediated. (A) Scheme of T cell transfer. Following EAU induction, $1 \times 10^7$ CD3⁺ T cells were enriched from inguinal lymph nodes and spleens of WT or *Alox5*⁻/⁻ mice and injected i.p. into *TCRb*⁻/⁻ mice. (B) Fundoscopy scores of WT or *Alox5*⁻/⁻ T cell-transferred *TCRb*⁻/⁻ recipients immunized with EAU, n = 6 mice per group. (C) Representative H and E histology of the posterior segment of the eye from *TCRb*⁻/⁻ recipients of WT or *Alox5*⁻/⁻ T cells, scale bar = 50 μm. (D, F) Flow cytometry analysis of total numbers of live cells and frequencies of CD4⁺ T cell populations producing IFN-γ or IL-17A in (D) eyes and (F) inguinal lymph nodes of WT and *Alox5*⁻/⁻ T cell recipients, n = 6 per group. (E, G) Flow

*Figure 5 continued on next page*

Figure 5 continued

cytometry analysis of chemokine receptor expression on activated CD4[+] T cells in (E) eyes and (G) inguinal lymph nodes of *TCRb*[-/-] recipients of WT and *Alox5*[-/-] T cells, n = 6 per group. *p<0.05, **p<0.01, Wilcoxon matched-pairs signed rank test and unpaired Welch's t test.
The online version of this article includes the following figure supplement(s) for figure 5:

**Figure supplement 1.** *Fpr2*[-/-] T cells amplify EAU pathogenesis.

mediator that is generated at sites of inflammation. Despite the established role of LXA$_4$ in innate immune cell regulation, direct regulation of adaptive immune responses by endogenous LXA$_4$ is not well defined in vivo, as most research efforts have focused on LXA$_4$'s ability to dampen and resolve acute inflammation elicited by innate immune cells. Beneficial anti-inflammatory effects of LXA$_4$ have been previously reported in the LPS-induced anterior uveitis model, however, pathogenesis of LPS-induced uveitis is driven exclusively by innate immune responses (*Karim et al., 2009*). In a murine asthma model induced by allergen immunization, LXA$_4$ treatment reduced airway inflammation by increasing natural killer cell-mediated eosinophil apoptosis (*Barnig et al., 2013*), demonstrating LXA$_4$ regulation of innate responses in vivo. Our data indicate that downregulated endogenous LXA$_4$, specifically in inguinal lymph nodes that drain the immunization site, is a key feature of

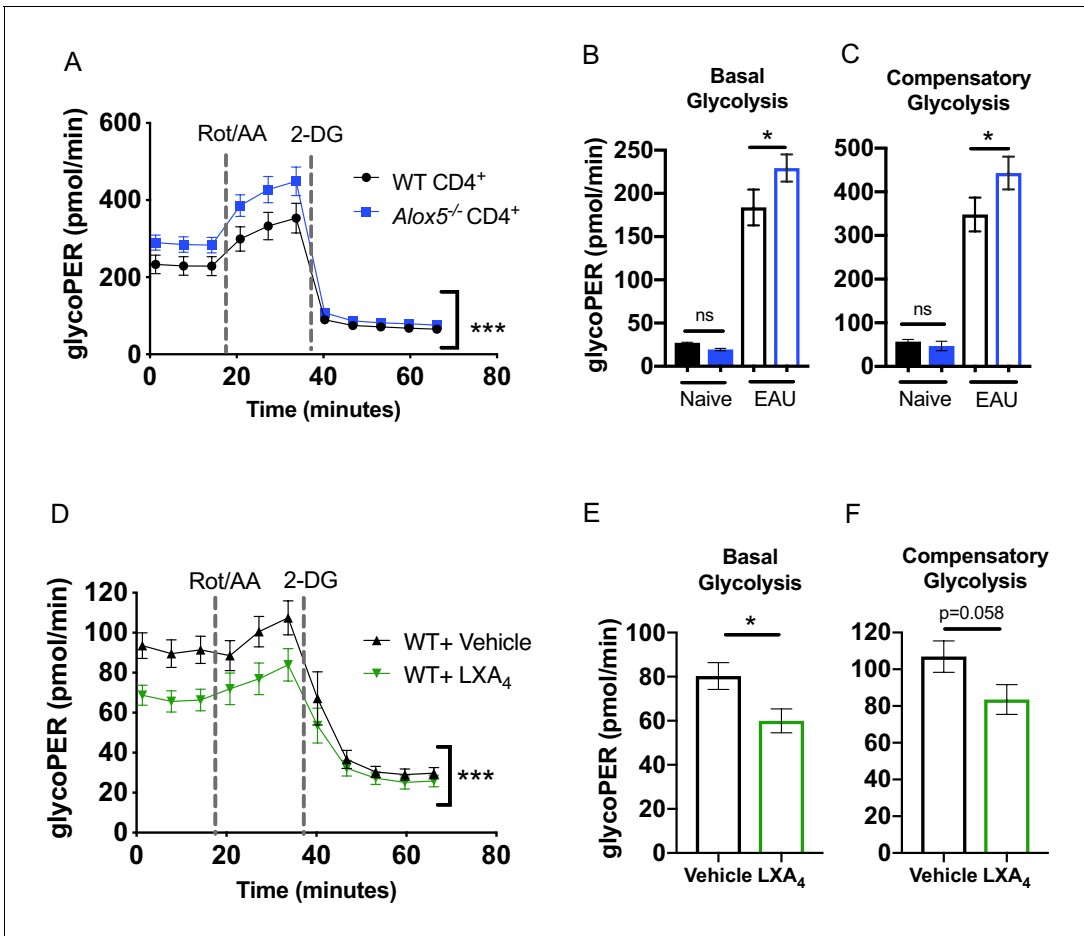

**Figure 6.** The LXA$_4$ pathway regulates effector function and metabolism of CD4[+] T cells. Inguinal lymph nodes and spleens were harvested from EAU-challenged WT and *Alox5*[-/-] mice on day 13 post-immunization. (A) CD4[+] T cells isolated from immunized WT and *Alox5*[-/-] mice were stimulated with anti-CD3 and anti-CD28 antibodies for 18 hr then subjected to glycolytic rate assay by Seahorse analyzer, n = 12 mice per group. (B–C) Bar graphs of (B) basal glycolysis and (C) compensatory glycolysis from the glycolytic rate assay in (A). (D) WT CD4[+] T cells isolated from immunized WT and *Alox5*[-/-] mice were pretreated with vehicle or LXA$_4$ in vitro and stimulated with anti-CD3 and anti-CD28 antibodies for 18 hr then subjected to glycolytic rate assay by Seahorse analyzer, n = 11 mice per group. (E–F) Bar graphs of (E) basal glycolysis and (F) compensatory glycolysis from the glycolytic rate assay in (D). *p<0.05, ***p<0.001, Wilcoxon matched-pairs signed rank test and Mann-Whitney test.

developing adaptive immunity, that is EAU. This is consistent with a previous report, which linked lower LXA$_4$ levels in draining lymph nodes to exacerbated dry eye disease in female mice (*Gao et al., 2015*). LXA$_4$-deficient *Alox5*$^{-/-}$ mice and adoptive transfer in T cell-deficient mice also demonstrated that in vivo disruption of the LXA$_4$ pathway amplifies effector T cell responses in auto-immune uveitis. These experiments provide strong evidence for LXA$_4$ regulation of adaptive immune responses initiated in lymph nodes by demonstrating that (a) LXA$_4$ deficiency causes dysregulated T effector cell function that leads to exacerbated autoimmune pathogenesis and (b) supplementation with exogenous LXA$_4$ reverses this effect. Thus, we identified a new role for LXA$_4$ in T cell-driven adaptive autoimmunity and uncovered a lymph node – eye axis that regulates this process.

It is not possible to capture changes in LXA$_4$ tissue levels after a low dose (1 µg) injection of LXA$_4$ at the time of tissue harvest since increases of LXA$_4$ in eye or lymph nodes are likely rapid and of short duration. LXA$_4$, like all eicosanoids, is enzymatically inactivated in minutes when not bound to an endogenous carrier protein such as albumin. However, several research groups have established that tail vein, intraperitoneal, subcutaneous or subconjunctival injections of LXA$_4$ or other SPMs (10 ng-1 µg) in mouse disease models induce long-lasting protective actions through changes in lympho-cyte and neutrophil function and macrophage phenotype at distant tissue sites (*Livne-Bar et al., 2017*; *Serhan et al., 2008*; *Gao et al., 2015*; *Fosshaug et al., 2019*; *Wei and Gronert, 2019*), which indicate direct actions at the target tissue.

Our findings provide evidence for LXA$_4$ regulation of T cell trafficking as a potential mechanism for controlling EAU progression. The S1PR1-sphingosine-1 phosphate axis is a primary mechanism for regulating lymph node T cell egress and lymphocyte recirculation. LXA$_4$ treatment selectively downregulated *S1pr1* expression only in inguinal lymph nodes of WT EAU-challenged mice. Pharma-cological downregulation of S1PR1 prevents trafficking of autoreactive T cells and is an FDA-approved treatment for multiple sclerosis (*Garris et al., 2014*). Hence, a role of the lymph node LXA$_4$ pathway in regulating *S1pr1* expression in lymphocytes warrants further investigation into the crosstalk between these two classes of lipid mediators. The CCR7-CCL19/CCL21 axis facilitates naïve T cell homing to secondary lymphoid organs, activated T cell egress, and the interaction between T cells and dendritic cells (DC) to induce peripheral tolerance or to elicit an adaptive immune response (*Comerford et al., 2013*). LXA$_4$ treatment differentially regulated this axis at the effector and initia-tion sites of the adaptive immune response by reducing *Ccr7* and *Ccl19* expression in the eye and increasing *Ccr7* and *Ccl21* expression in inguinal lymph nodes.

The endogenous role of LXA$_4$ in regulating effector T cell function through *Ccr7* was confirmed with *Alox5*$^{-/-}$ mice using T cell adoptive transfer and in vitro migration experiments. LXA$_4$ deficiency was coincident with dysregulated (i.e. lower) expression of CCR7 in T cells from inguinal lymph nodes, increased IFN-γ production and an almost two-fold increase of CD4$^+$ T cell migration towards CCL19/21 gradient. *Ccr7* is expressed predominantly in DC and T cells, and its ligands CCL19 and CCL21 can be secreted by several cell types. In this study, we did not investigate the cellular source of CCL19 and CCL21 in tissues, the effect of LXA$_4$ on *Ccr7* expression in DC or assess direct LXA$_4$ regulation of *Ccr7* expression on T cells. LXA$_4$ has established actions with DC and other cell types, therefore, we cannot rule out that LXA$_4$ also controls T cell responses in inguinal lymph nodes through priming by DC and/or regulation of CCL19 and CCL21 production by other cell types.

T cell glycolysis is vital in regulating T cell effector function. Metabolic switch from oxidative phos-phorylation to glycolysis is required to sustain T effector function and cytokine production (*Coe et al., 2014*; *Salmond, 2018*). Inhibition of glycolysis abrogates IFN-γ production in activated CD4$^+$ T cells, implicating glycolysis as regulator of sustained T effector responses (*Chang et al., 2013*). LXA$_4$ treatment decreased CD4$^+$ T cells glycolytic responses and reduced IFN-γ production from CD4$^+$ T cells, providing evidence of direct regulation of T cell function by LXA$_4$. Increased gly-colytic responses of CD4$^+$ T cells from *Alox5*$^{-/-}$ mice underscores the relevance of lymph node resi-dent LXA$_4$ in controlling T cell effector function. T cell glycolysis and mitochondrial metabolism as potential therapeutic targets are supported by elevated glycolytic metabolism in T cells of systemic lupus erythematosus (SLE) patients and reduced IFN-γ production and disease biomarkers upon inhi-bition of CD4$^+$ T cell glycolysis in mouse models of SLE (*Yin et al., 2015*). The ability of LXA$_4$ to downregulate glycolysis in activated CD4$^+$ T cell is of interest and suggests that the lymph node resi-dent LXA$_4$ pathway is a potential checkpoint to limit inflammatory cytokine production and the mag-nitude of adaptive immune responses.

In mice and humans, $LXA_4$ cognate receptor ALX/FPR2 is expressed in T cells, B cells and innate lymphoid cells (*Wei and Gronert, 2019*), and has been shown to regulate $CD4^+$ T cell cytokine production in vitro (*Chiurchiù et al., 2016*). During peak EAU, *Fpr2* expression in $CD4^+$ T cells was highly upregulated in the inguinal lymph nodes that drain the site of immunization, suggesting that increased capacity of $LXA_4$ signaling in T cells is a counter-regulatory mechanism in response to the development of adaptive immunity. Consistent with our hypotheses that dysregulated $LXA_4$ signaling allows for development of EAU and facilitates augmented effector T cell functional responses, adoptive transfer of either $Alox5^{-/-}$ or $Fpr2^{-/-}$ T cells to $TCRb^{-/-}$ recipients caused exacerbated EAU (*Figure 5—figure supplement 1A*). Transferred T cells from $Fpr2^{-/-}$ and $Alox5^{-/-}$ mice induced different signatures of inflammatory ocular infiltrates (*Figure 5—figure supplement 1B and C*), likely due to differential regulation of donor T cells and innate effector cells by the host $LXA_4$ pathway in $TCRb^{-/-}$ recipient mice. However, unlike $Alox5^{-/-}$ mice, global $Fpr2^{-/-}$ mice (which generate $LXA_4$) did not develop more severe EAU than WT controls after immunization (*Figure 5—figure supplement 1D*). A second $LXA_4$ receptor GPR32 has been identified in humans (*Dalli and Serhan, 2019*), which currently has no mouse homolog. Our data suggest that additional receptors for $LXA_4$ may exist in mice. Incomplete abrogation of $LXA_4$ signaling due to a putative second receptor may explain why immunized global $Fpr2^{-/-}$ mice did not develop more severe autoimmune disease.

EAU is an induced and well characterized model of autoimmune uveitis that recapitulates features of human posterior autoimmune uveitis. Although the immune-regulatory roles of $LXA_4$ in human draining lymph nodes remain to be defined, $LXA_4$ has been identified as an abundant SPM in human axillary lymph nodes (*Colas et al., 2014*). While it is important to note that there are significant differences in mouse and human immune responses (*Seok et al., 2013*), $LXA_4$ pathway and receptors are conserved in humans and mice and are targeted as novel therapies based on promising results from animal studies.

In summary, our data established that the basal tone of $LXA_4$ in lymph nodes is selectively downregulated in the draining lymph node during development of an adaptive immune response. We identified regulation of CCR7 and S1PR1 expression and glycolytic responses in $CD4^+$ T cells as new mechanisms for $LXA_4$ to control effector T cell functions and egress from lymph nodes, which establishes $LXA_4$ as an important homeostatic regulator for fine tuning adaptive immune responses. Thus, amplification of the resident lymph node $LXA_4$ pathway may be a potential target to prevent the development of pathogenic adaptive immune responses and autoimmunity.

## Materials and methods

### Patients and healthy controls

Male and female patients ages 30–76 with clinical diagnosis of non-infectious posterior segment uveitis were enrolled in the National Eye Institute protocol number 16-EI-0046. Healthy controls were NIH blood bank donors of both sexes with a similar age range. Serum samples were obtained from male and female patients ages 30–76 with clinical diagnosis of non-infectious posterior uveitis, healthy controls were NIH blood bank donors of both sexes with a similar age range whose samples were de-identified and sent to the lab. Patients were enrolled from May 2017 to July 2018 under a clinical research protocol (NCT02656381), approved by the institutional review board of the National Institutes of Health. Informed consent (including publishing language as required by NIH IRB) were obtained from all subjects. The study adhered to the tenets of the Declaration of Helsinki.

### Serum ELISA assay

Serum $LXA_4$ levels were measured by ELISA (Neogen, KY) per manufacturer instruction and read on SpectraMax iD3 microplate reader (Molecular Devices, CA).

### Mice, EAU induction and $LXA_4$ treatment

C57BL/6J mice between 8 and 12 weeks of age were purchased from the Jackson Laboratory and were used as the WT strain. All experimental procedures were approved by the Animal Care and Use Committee at University of California, Berkeley (protocol AUP-2016-04-8691-1), and the National Eye Institute at the National Institutes of Health (animal protocol NEI-581) and are in accordance with the National Institutes of Health Guide for the Care and Use of Laboratory Animals. All

animals were maintained on the NIH-31 Open Formula diet: 4.7% Fat, 1.2% monounsaturated fatty acids, 2.1% polyunsaturated fatty acids, with 1.9% C18:2 ω6 linoleic acid and 0.2% C18:3 ω3 linolenic acid (Envigo, WI), starting at 3 weeks of age through the duration of the experiments. For induction of EAU, each mouse was immunized with 150 or 300 μg IRBP$_{651-670}$ peptide (Genscript, NJ or BioBasic, NY) as indicated, emulsified in CFA supplemented with 2.5 mg/ml *Mycobacterium tuberculosis* (Sigma, MO), and injected with 1 μg pertussis toxin (Sigma, MO) as reported earlier (*Mattapallil et al., 2015*). *Alox5$^{-/-}$* mice (B6.129S2-*Alox5$^{tm1Fun}$*/J, stock number 004155) and *TCRb$^{-/-}$* mice (B6.129P2-*Tcrb$^{tm1Mom}$*/J, stock number 002118) were purchased from the Jackson Laboratory. *Fpr2$^{-/-}$* mice (*Dufton et al., 2010*) were gifted by Dr. Asma Nusrat from the University of Michigan. For assessing therapeutic effects of LXA$_4$, either 1 μg/100 μl of LXA$_4$ (Cayman Chemical, MI), which was resolved in 100% EtOH and diluted to 10% in PBS, or 100 μl of 10% EtOH in PBS (vehicle control) was injected subcutaneously and intraperitoneally starting at day 0 of disease induction. IRBP-specific TCR transgenic R161H mice (*Horai et al., 2013*) generated on the congenic CD90.1 B10.RIII background and WT CD90.2 B10.RIII mice (Jackson Laboratory, ME) were maintained in house at the NIH animal facility.

## Evaluation of EAU by fundoscopy and histology

Funduscopic scoring system was used to evaluate activity of inflammation in the retina. Mice were anesthetized with ketamine and xylazine (7:3) and the retinas were examined under a binocular microscope after pupil dilation with tropicamide and phenylephrine. Scores ranging from 0 to 4 were assigned to each eye according to the extent and severity of inflammatory lesions (*Agarwal et al., 2012*). For histological evaluation, eyes were enucleated, pre-fixed in 4% glutaraldehyde for 1 hr, post-fixed in 10% buffered formalin, embedded in paraffin and processed for H and E staining. Scores were determined in a blinded manner on a scale of 0–4 in half-point increments according to published EAU scoring criteria that are based on the type, size, severity and number of lesions (*Caspi, 2003*).

## Electroretinography and fundus imaging

Mice were dark-adapted overnight, and experiments were performed under dim red illumination. Electroretinograms were recorded using Espion E2 System (Diagnosys LLC, MA) that generated and controlled light stimuli at 10 cds/m$^2$ for a-wave and b-wave (*Chen et al., 2013*). A reference electrode was inserted above the tongue, corneal electrodes were placed at the center of corneas with sterile lubricant eye gel applied to the corneal electrodes. The amplitudes of a-wave and b-wave were analyzed and measured using Espion software. Images was captured using Micron III retinal imaging microscope (Phoenix Research Labs, CA) for small rodents.

## Tissue harvest and processing

Eyes, lymph nodes and spleens were harvested from vehicle and LXA$_4$-treated immunized mice 16 days post-immunization after extensive perfusion through the heart. Enucleated eyes were trimmed of lenses and connective tissues, minced and treated with 1 mg/ml collagenase D for 40 min at 37°C. Lymph nodes and spleens were mashed, cells filtered through 40 μm strainers and resuspended to make single cell suspensions. Red blood cells in spleens were lysed with ACK lysis buffer. Whole blood was allowed to clot for 30 min at room temperature. Samples were centrifuged and serum supernatant collected for analysis.

## Real-time quantitative PCR

RNA was isolated from mouse tissues using RNeasy isolation kit (Qiagen, Germany), and reverse transcribed with High Capacity cDNA kit (Applied Biosystems, CA). Mouse β-actin was used as the endogenous reference gene. PCR was performed using Taqman gene expression assays (Thermo Fisher Scientific, MA) with following primer/probe sets. *Alox5*: Mm01182747_m1, *Alox15*: Mm00507789_m1, *Fpr2*: Mm00484464_s1, *S1pr1*: Mm00514644_m1.

## Detection of endogenous formation of eicosanoids

Whole lymph nodes and eye globes were harvested at disease onset and peak inflammation from mice induced for EAU by immunization with IRBP$_{651-670}$ peptide. Tissues from naïve mice served as

healthy controls. Tissues were homogenized in 66% MeOH containing deuterated internal standards with Bead Ruptor, and lipid mediators were extracted using solid phase C-18 columns. Mouse serum was combined with 2 volumes of ice cold MeOH containing deuterated internal standards. Diluted serum samples were vortexed and incubated at −80°C for 1 hr to precipitate proteins. Supernatants were collected and extracted using solid phase C-18 columns (*Gao et al., 2018*). Lipid mediators were then quantified by liquid chromatography-tandem mass spectrometry (LC-MS/MS). Extraction recovery was calculated based on deuterated internal standards ($PGE_2$-d4, $LTB_4$-d4, 15-HETE-d8, $LXA_4$-d5, DHA-d5, AA-d8). The LC-MS/MS system was composed of Agilent 1200 series HPLC, Kinetex C18 minibore column (Phenomenex, CA), and AB Sciex QTRAP 4500 mass spectrometer (SCIEX, MA). Analyses were performed in negative ion mode with scheduled multiple reaction monitoring using 4–5 transition ions per lipid mediator (*Livne-Bar et al., 2017*; *von Moltke et al., 2012*).

## Flow cytometry analyses of T cells

Single cell suspensions were prepared from isolated tissues. Samples of $1 \times 10^6$ cells were stained with anti-mouse antibodies conjugated with following fluorochromes: BV421, BV510/V500, FITC, PE, PerCP-Cy5.5/PerCP-eF710, PE-Cy7, APC, APC Cy7/APC-eF780, BV650/SB645, BV785. Surface staining was performed using antibodies against F4/80, Ly6G, CD3, CD4, CD11b, CD44, CCR6, CXCR3, CCR7, and S1PR1. Ghost Dye UV 450 or Red 780 (Tonbo, CA) were used to exclude dead cells. Intracellular staining was performed after stimulating cells with 50 ng/ml PMA and 1 µg/ml Ionomycin (MilliporeSigma, MA) in the presence of Brefeldin A (Golgi Plug, BD Biosciences, CA) for 4–6 hr at 37°C in cell culture media. Following the surface and viability staining, cells were fixed with 4% paraformaldehyde (PFA) and permeabilized with Triton buffer (0.5% Triton X-100% and 0.1% BSA in PBS). Fluorochrome-conjugated (see above) anti-mouse Foxp3, IL-17A, and IFN-γ were used for intracellular staining. Cells were acquired on BD LSR Fortessa (BD Biosciences, CA), MACSQuant Analyzer (Miltenyi Biotec, Germany), or Cytoflex (Beckham Coulter, CA). Fcs files were analyzed using FlowJo (FlowJo LLC, OR).

## Immunohistochemistry

Tissues were embedded in OCT and cryosectioned into 7 µm sections. Tissues sections were fixed in 4% PFA and blocked with serum prior to primary antibody staining. Primary antibodies against mouse CD4 (Biolegend, CA) and COX-2 (Abcam, UK) were used. Anti-goat AlexaFluor 488 and AlexaFluor 568 (ThermoFisher Scientific, MA) secondary antibodies were used for immunofluorescent labeling. Cells were counterstained with DAPI at 1:3000 dilution (Sigma, MO). Fluorescent images were acquired with a Zeiss AxioImager microscope at 10X and 20X.

## Quantitation of immunohistochemistry images

Images were quantified in Fiji software. Particle mean intensity was measured in each fluorescent channel of retina and choroid tissues. Statistical significance was calculated on 3–7 mice per experimental group.

## T cell proliferation assays

$5 \times 10^5$ lymph node cells resuspended in HL-1 media (Lonza, GA) were plated in 96-well round bottom plates and stimulated with serial dilutions of $IRBP_{651-670}$ peptide. Cells were cultured for 2 days and 1 µCi/well of $[^3H]$ thymidine was added and incubated for 16 hr. Samples were harvested and counted by liquid scintillation (Perkin Elmer, MA).

## Adoptive transfer of retina-specific T cells and LXA₄ treatment

To elicit EAU by adoptive T cell transfer, $5 \times 10^6$ $IRBP_{161-180}$ in vitro-activated (2 µg/ml for 3 days) lymph node cells from R161H CD90.1 mice were injected intraperitoneally into B10.RIII WT CD90.2 mice. $LXA_4$ was administered at the doses of 1 µg or 250 ng every day throughout the disease course. Fundoscopy was assessed starting on day four after cell transfer to track disease development. Alternatively, donor C57BL/6 WT and $Alox5^{-/-}$ mice were immunized with $IRBP_{651-670}$ peptide, and $CD3^+$ T cells purified with T cell enrichment columns (R and D Systems, MN) from lymph nodes that drain immunization sites and spleens on day 11 post-immunization. WT or $Alox5^{-/-}$ T cells ($1 \times 10^7$ each) were transferred intraperitoneally into $TCRb^{-/-}$ recipients. Three weeks after the cell

transfer, *TCRb*[-/-] recipient mice were immunized with 150 µg IRBP$_{651-670}$ in CFA and pertussis toxin as described above.

## NanoString assays for transcriptomic analysis

RNA was extracted from snap frozen tissues using RNeasy (Qiagen, Germany) and analyzed by nCounter Analysis Technology using the mouse cancer immunology panel (nanoString, WA). Data were normalized to housekeeping genes and analyzed using nSolver software. Gene expression threshold was set at 30 counts to eliminate background signals.

## Transwell migration assay and in vitro T cell assay

CD4$^+$ T cells were isolated using CD4 T cell isolation kit (STEMCELL Technologies, Canada) from inguinal lymph nodes and spleens of mice immunized with IRBP$_{651-670}$ and stimulated in vitro with 2 µg of anti-CD3 and 1 µg of anti-CD28 antibodies for 18 hr. Stimulated T cells were seeded at $2 \times 10^5$ per transwell insert (Corning, MA) and allowed to migrate toward 100 ng of CCL19 and 100 ng of CCL21 chemokine ligands (R and D systems, MN) in 2% FCS for 4 hr. 10 nM – 2 µM of LXA$_4$ was added to stimulated T cells cultured in HL-1 media supplemented with 0.5% normal mouse serum and stained for chemokine receptors. Cells were analyzed on Cytoflex.

## Seahorse metabolic assays

Seahorse XFe96 Extracellular Flux Analyzer (Agilent Technologies, CA) was used to measure glycolysis. CD4$^+$ T cells were harvested from WT or *Alox5*[-/-] mice immunized with IRBP$_{651-670}$ and stimulated with anti-CD3 and anti-CD28 antibodies for 18 hr. The glycolysis proton efflux rate (glycoPER) for each well was calculated from cells subjected to XF Glycolytic Rate assays with the following concentrations of injected compounds: 50 mM 2-DG and 0.5 µM rotenone/antimycin A. For assays with LXA$_4$ treatment, CD4$^+$ T cells were preincubated with 1 µM of LXA$_4$ or vehicle control for 1 hr before anti-CD3 and anti-CD28 stimulation.

## Statistical analysis

Statistics were performed using GraphPad Prism 8 (GraphPad, CA). Statistical differences were determined by ANOVA, Welch's t-tests, and Wilcoxon matched-pairs signed rank test for parametric variables and Mann-Whitney test for nonparametric variables. All error bars are standard error mean. p value < 0.05 was considered statistically significant and is denoted with *p≤0.05, **p≤0.01, ***p≤0.001 in the figures.

## Acknowledgements

Funding: This work was supported by grants EY026082 to KG and EY000184 NEI Intramural funding to RRC. Competing interests: The authors declare no financial interests. Data and materials availability: All data needed to evaluate the conclusions of the paper are present in the main text and Supplementary Materials. The authors thank Ryan Salvador and Jihong Tang for assistance in tissue harvest, Amy Zhang for advice on statistical analyses, Benjamin Smith for developing the fluorescence quantification program in Fiji, Julie Laux, Rafael Villasmil, Arvydas Maminishkis, Jennifer Kielczewski, Phyllis Silver, Allison Chan, Victoria Ly, Nicole Rossi, Kimberly Tang, Julia Yoo and Karthik Mouli for technical support, and members of the Caspi lab for helpful discussions.

## Additional information

### Funding

| Funder | Grant reference number | Author |
| --- | --- | --- |
| National Eye Institute | EY026082 | Karsten Gronert |
| National Eye Institute | EY000184 | Rachel R Caspi |

The funders had no role in study design, data collection and interpretation, or the decision to submit the work for publication.

## Author contributions
Jessica Wei, Conceptualization, Formal analysis, Investigation; Mary J Mattapallil, Resources, Formal analysis, Investigation; Reiko Horai, Formal analysis, Investigation, Methodology; Yingyos Jittayaso-thorn, Arnav P Modi, Investigation, Methodology; H Nida Sen, Resources; Karsten Gronert, Rachel R Caspi, Conceptualization, Resources, Supervision, Funding acquisition, Project administration

## Author ORCIDs
Jessica Wei (iD) http://orcid.org/0000-0002-7329-2812
Karsten Gronert (iD) https://orcid.org/0000-0002-5329-7907

## Ethics
Human subjects: Male and female patients ages 30 - 76 with clinical diagnosis of non-infectious posterior segment uveitis were enrolled in the National Eye Institute protocol number 16-EI-0046. Healthy controls were NIH blood bank donors of both sexes with a similar age range. Serum samples were obtained from male and female patients ages 30 - 76 with clinical diagnosis of non-infectious posterior uveitis, healthy controls were NIH blood bank donors of both sexes with a similar age range whose samples were de-identified and sent to the lab. Patients were enrolled from May 2017 to July 2018 under a clinical research protocol 428 (NCT02656381), approved by the institutional review board of the National Institutes of Health. Informed consent (including publishing language as required by NIH IRB) were obtained from all subjects. The study adhered to the tenets of the Declaration of Helsinki.

Animal experimentation: All experimental procedures were approved by the Animal Care and Use program at University of California, Berkeley, and the National Eye Institute at the National Institutes of Health.(protocols AUP-2016-04-8691-1 and NEI-581).

## Decision letter and Author response
Decision letter https://doi.org/10.7554/eLife.51102.sa1
Author response https://doi.org/10.7554/eLife.51102.sa2

## Additional files

### Supplementary files
- Supplementary file 1. Key Resources Table.
- Transparent reporting form

### Data availability
All data needed to evaluate the conclusions of the paper are present in the main text and supplementary materials.

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
