## [Decision Letter]

**Acceptance summary:**

In experimental (mouse) autoimmune uveitis the authors found a basal level of the lipid mediator LXA_4_ in lymph nodes that decreases in the draining lymph node during adaptive immune responses. LXA_4_, a Specialized Pro-resolving Mediator, has a clinically relevant role in regulating acute inflammation and inflammatory resolution.

**Decision letter after peer review:**

Thank you for submitting your article "A novel role for lipoxin A_4_ in driving a lymph node-eye axis that controls autoimmunity to the neuroretina" for consideration by *eLife*. Your article has been reviewed by three peer reviewers, one of whom is a member of our Board of Reviewing Editors, and the evaluation has been overseen by Tadatsugu Taniguchi as the Senior Editor. The following individual involved in review of your submission has agreed to reveal their identity: Nicolas G Bazan (Reviewer #2).

The reviewers have discussed the reviews with one another and the Reviewing Editor has drafted this decision to help you prepare a revised submission. Please aim to submit the revised version within two months.

Summary:

Wei et al. conducted experiments to determine whether and how LXA_4_ regulates T effector cell function in the pathogenesis of autoimmune uveitis. They showed that LXA_4_ serum concentration may be increased in posterior uveitis patients, and the disease severity can be altered by modulating LXA_4_ concentration, i.e. ALOX5-deficient mice and LXA_4_ injection in the mouse experimental autoimmune uveitis (EAU) model.

These findings clearly provide evidence for the involvement of LXA_4_ signaling pathways in the pathogenesis of autoimmune uveitis. However, the study falls a little short to provide sufficient mechanistic insight how LXA_4_ regulates T effector cell function in the pathogenesis of EAU. The main criticism is that it is not clear where is the specific site of LXA_4_ action to regulate T cell function. They suggest that the resident LXA_4_ circuit in inguinal lymph node plays roles, but it is not clear if the LXA_4_ signaling is up-regulated or down-regulated in the inguinal lymph node during the EAU, since LXA_4_ concentration goes down while All/Fpr2 expression goes up (Figure 1C and 1E). After LXA_4_ i.p. injections, LXA_4_ concentration should be determined in eyes, inguinal lymph, and serum. 5-LOX is involved in the production of not only LXA_4_, but also leukotriene species. Also, the Cox pathway might be augmented in the compensatory manner in the 5-LOX-deficient background. So it is not clear whether the EAU phenotype observed in the ALOX4^-/-^ mice can be attributed to LXA_4_ alone or not. In the adoptive transfer experiment of T cells, they transfer the T cells from the EAU mice. These T cells are already activated in the global ALOX5-/- environment, and do not support the conclusion that effect of LXA_4_ is T cell mediated.

As potential mechanisms, they suggest the regulations by LXA_4_ of chemokine receptors and sphingosine-1 receptor expression as well as glucose metabolism. These observations are descriptive, only based on the gene expression levels and glycolytic rate, not providing the direct regulation by LXA_4_.

Essential revisions:

1) Figure 1A.

It is important to see if the rodent findings translate to human inflammation / immune responses. So this figure is critical. Within the "other" uveitis group was there a difference between "pan uveitis" which also has posterior (choroidal) uveitis and "anterior uveitis"? Anterior uveitis alone may be a better control if you suggest that specific ocular tissue inflammation determines the level of LXA_4_ and thus pan uveitis should be added to the posterior group. If on the other hand you suggest the instigating insult is associated with LXA_4_ levels then more explanation about the choices of categories of human disease is needed particularly because the difference between groups you currently have defined is small. An unbiased approach (put all human data together and see which patient groups are associated with what LXA_4_ levels) would be the most telling. "Dot plots" should be done as in the rest of the paper as it would give more information.

2) There are great differences between mouse and human inflammation/immune responses (Seok et al., 2013). Please comment in detail in the Discussion and Introduction with respect to this work.

3) Figure 2.

Alox5^-/-^ mice lack more than LXA_4_. What are the possible effects of other metabolic products that are missing in these mice?

What was the diet composition for the mice specifically regarding omega 3 and 6 LCPUFAs and was it maintained throughout as standard chow can vary batch to batch? A diet high in AA versus DHA would likely influence results.

Figure 2D The authors are presenting a single point for the electroretinography analysis, for which we have no indication on the light intensity used to get this b-wave response. In Materials and methods it is mentioned that the ERG was performed with controlled light stimuli (subsection “Electroretinography and fundus imaging”), meaning that more than one stimulus was performed, however this figure presents the response from only one intensity point. Moreover, the aforementioned subsection mentions a-wave, which are not apparent in this figure, nor discussed in Results. A complete ERG plot with increasing light stimuli showing a-wave and b-wave would be more appropriate.

4) Figure 3.

Figure 2G and Figure 3H Are the statistical analysis missing in these graphs or the differences are not significant? If the latter mention it in the legend.

Figure 3C What is the mouse strain background of EAU and EAU+LXA_4_ animals, because their RPE is not pigmented. This is not clear, because according to the subsection “Mice, EAU induction and LXA_4_ treatment”, C57BL/6J (pigmented animals) were used for the EAU induction and LXA_4_ treatment. Can the authors clarify the reason of the lack of pigmentation in the EAU and EAU+LXA_4_ pictures?

Figure 3D Can the authors improve the quality of their immunofluorescent pictures? For example, the DAPI staining is barely visible, which is essential for the identification of the layering of the retina and proper morphological localization of the labeled targets. Moreover, a thicker line for the scale bar to make it more visible would help improve this picture.

5) Mechanism.

As potential mechanisms, they suggest the regulations by LXA_4_ of chemokine receptors and sphingosine-1 receptor expression as well as glucose metabolism. These observations are descriptive, only based on the gene expression levels and glycolytic rate, not providing the direct regulation by LXA_4_.

To strengthen their conclusion, additional experiments listed below are needed.

1) Provide LXA_4_ concentrations in the eye, inguinal lymph node, and serum after i.p. injection of LXA_4_ during the EAU model.

2) Provide evidence or discussion to show that other lipid mediators (i.e. leukotrienes and prostaglandins) are not affected in the ALOX5-deficient mice.

3) Examine if LXA_4_ treatment of isolated T cells in vitro can impact the gene expression levels of chemokine receptors and S1pr1, like they did in the glycolysis analysis (Figure 6).

---

## [Author Response]

Essential revisions:1) Figure 1A.It is important to see if the rodent findings translate to human inflammation / immune responses. So this figure is critical. Within the "other" uveitis group was there a difference between "pan uveitis" which also has posterior (choroidal) uveitis and "anterior uveitis"? Anterior uveitis alone may be a better control if you suggest that specific ocular tissue inflammation determines the level of LXA4 and thus pan uveitis should be added to the posterior group. If on the other hand you suggest the instigating insult is associated with LXA4 levels then more explanation about the choices of categories of human disease is needed particularly because the difference between groups you currently have defined is small. An unbiased approach (put all human data together and see which patient groups are associated with what LXA4 levels) would be the most telling. "Dot plots" should be done as in the rest of the paper as it would give more information.

As the reviewers suggested we reanalyzed the human serum data. Detailed analysis of all uveitis groups identified patients with 13 distinct diagnoses. Only 1 patient had a diagnosis of anterior uveitis. All other uveitides can include posterior uveitis and/or immune responses as part of the pathogenesis. In total there were 4 uveitis variants with only 1 patient (edema, retinal detachment, anterior uveitis, photopsia), which were excluded from the new analysis. All other uveitides ranged from 2 to 23 patient serum samples per group. Decoding of the samples also revealed that our original dataset erroneously included a number of plasma (and not serum) samples, which skewed the mean of the uveitis groups. After excluding the plasma samples, the new analysis revealed an opposite result from our original finding, namely serum LXA_4_ levels in patients with posterior uveitis symptoms (n=78) were significantly lower (p=0.0092) than in healthy controls (n=41). Revised Figure 1A shows significantly lower serum levels of LXA_4_ when all posterior uveitis patients were compared to healthy controls. All individual diagnosis groups had means that were lower than the healthy controls, with the exception of Vogt-Koyanagi-Harada disease (VKH, n=9). Due to the variation in sample size in each group, differences between individual diagnosis groups and healthy controls did not reach significance, except for intermediate uveitis (n=4, p=0.04). We are fortunate to have access to these human samples, but it is a post-hoc analysis of all available serum samples and we cannot increase the number of samples for individual diagnosis groups to establish which type of uveitis or pathogenesis correlates with the most significant decrease in serum LXA_4_ levels. Our findings add to a growing body of evidence that has recently been published by the Serhan and Dalli research groups (1-4), which document dynamic and disease-specific changes in SPM serum or plasma levels in humans. A new section for Figure 1A (subsection “LXA_4_ is generated during autoimmune uveitis in a time- and site- dependent manner”) has been added to Results and corresponding discussion (Discussion, first paragraph) has been added.

2) There are great difference between mouse and human inflammation/ immune responses (Seok et al., 2013). Please comment in detail in the Discussion and Introduction with respect to this work.

We fully agree that there are differences in mouse and human inflammation and immune responses and that all findings from mouse models have to eventually be validated in humans. As requested by the reviewer we added the PNAS reference that compared genomic responses in mice and humans and added discussion to highlight the limitations of our mouse model (Discussion, seventh paragraph). It is important point out that despite limitations of mouse models, major breakthroughs in our understanding of human immunology and disease treatment were discovered in mouse and invertebrate models (e.g., 2011 and 2018 Nobel Prize in Physiology or Medicine).

3) Figure 2.Alox5-/- mice lack more than LXA4. What are the possible effects of other metabolic products that are missing in these mice?

5-LOX is also the required enzyme for generation leukotrienes. LTB_4_ and peptido-leukotriene (LTC_4_/D_4_/E_4_) formation is impaired in Alox5^-/-^ mice. LTB_4_ has established roles in regulation of innate immune cells and Th2 responses. However, in the Th1/17 -driven autoimmune uveitis model, peptido-leukotrienes were not consistently detected in tissues and LTB_4_ levels did not significantly change in lymph nodes or the eye during EAU development (new Figure 2—figure supplement 1), which is why our study focused on the 5-LOX metabolite LXA_4_. Our previous studies (5, 6), which investigated SPM formation in draining lymph nodes during the initiation of ocular surface immune responses, also did not detect significant levels or changes in LTB_4_. 5-LOX is also the required enzyme for the formation of several DHA-, DPA- and EPA-derived SPM; however, these were not consistently detected in lymph nodes or the eye with our LC-MS/MS method. New text has been added to the Results section (subsection “LXA_4_ is generated during autoimmune uveitis in a time- and site- dependent manner”, second paragraph) to justify our focus on LXA_4_ as the most relevant 5-LOX metabolite in the EAU model.

What was the diet composition for the mice specifically regarding omega 3 and 6 LCPUFAs and was it maintained throughout as standard chow can vary batch to batch? A diet high in AA versus DHA would likely influence results.

All animals were maintained on the NIH-31 Open Formula diet with 1.9% C18:2 ω6 linoleic acid and 0.2% C18:3 ω3 linolenic acid (Envigo, WI). Vendor, name and PUFA composition of the standard rodent chow have been added to the Materials and methods section. Mice were housed in the same animal room and experienced the same controlled environmental conditions. Control and experimental animals were on the same open formula NIH rodent diet from weaning through the duration of the experiment thus batch to batch variation in the diet would equally affect all experimental and control groups. We agree that batch to batch variation in rodent diets could potentially change the ratio of dietary intake of AA:DHA:EPA. However, in all our tissue lipidomic analyses AA, DHA and EPA are included as standard analytes and we did not notice any marked changes in the ratio of AA:DHA:AA among different sets of mouse experiments.

Figure 2D The authors are presenting a single point for the electroretinography analysis, for which we have no indication on the light intensity used to get this b-wave response. In Materials and methods it is mentioned that the ERG was performed with controlled light stimuli (subsection “Electroretinography and fundus imaging”), meaning that more than one stimulus was performed, however this figure presents the response from only one intensity point. Moreover, the aforementioned subsection mentions a-wave, which are not apparent in this figure, nor discussed in the Results. A complete ERG plot with increasing light stimuli showing a-wave and b-wave would be more appropriate.

Only one light intensity was used per condition. Light stimulus of 10cds/m2 was used for a-wave and b-wave. Details for light stimuli have been added to the Materials and methods section and a-wave data has been added to revised Figure 2D.

4) Figure 3.Figure 2G and Figure 3H Are the statistical analysis missing in these graphs or the differences are not significant? If the latter mention it in the legend.

No statistical analyses were performed in these graphs because error bars depict n=3 technical replicates from pooled n=6-9 animals per experimental group. Using ANOVA, stimulation index of proliferation assays in 2G and 3H show statistical significance between EAU vs. EAU+ LXA_4_ groups, as well as WT vs. Alox5^-/-^ groups (Author response image 1). Showing both stimulation index as well as cpm in the manuscript would be redundant, therefore we are appending the figures in the rebuttal only.

**Author response image 1. respfig1:** Stimulation index of proliferation assays from A) vehicle vs LXA_4_ treated EAU-challenged mice, p= 0.0154 and B) WT vs Alox5^-/-^ EAU-challenged mice, p= 0.0248. Two-way ANOVA.

Figure 3C What is the mouse strain background of EAU and EAU+LXA4 animals, because their RPE is not pigmented. This is not clear, because according to the subsection “Mice, EAU induction and LXA4 treatment”, C57BL/6J (pigmented animals) were used for the EAU induction and LXA4 treatment. Can the authors clarify the reason of the lack of pigmentation in the EAU and EAU+LXA4 pictures?

The mouse strain for all experiments is C57BL6, lack of pigmentation was due to poor imaging from an old microscope. Slides are now digitally scanned in high resolution and images have been replaced.

Figure 3D Can the authors improve the quality of their immunofluorescent pictures? For example, the DAPI staining is barely visible, which is essential for the identification of the layering of the retina and proper morphological localization of the labeled targets. Moreover, a thicker line for the scale bar to make it more visible would help improve this picture.

Quality of immunofluorescent images has been improved. DAPI fluorescence was enhanced and scale bar thickened in images.

5) Mechanism.As potential mechanisms, they suggest the regulations by LXA4 of chemokine receptors and sphingosine-1 receptor expression as well as glucose metabolism. These observations are descriptive, only based on the gene expression levels and glycolytic rate, not providing the direct regulation by LXA4.To strengthen their conclusion, additional experiments listed below are needed.1) Provide LXA4 concentrations in the eye, inguinal lymph node, and serum after i.p. injection of LXA4 during the EAU model.

It is not possible to capture changes in LXA_4_ tissue levels after a low dose (1 μg) injection of LXA_4_ at the time of tissue harvest since, increases in eye or lymph node LXA_4_ levels are likely rapid and of short-duration (minutes). LXA_4_, like all eicosanoids, is enzymatically inactivated in minutes when not bound to an endogenous carrier protein such as albumin. However, several research groups have established that tail vein, intraperitoneal, subcutaneous or subconjunctival injections of LXA_4_ or other SPMs (10 ng-1 mg) in mouse disease models induce long-lasting protective actions through changes in lymphocyte and PMN function and macrophage phenotype at distant tissue sites. We did measure LXA_4_ in inguinal lymph nodes in LXA_4_ or vehicle control treated mice on day 10 and day 16 in the pathogenesis of EAU (Author response image 2). Since the tissue was collected 2-3 hours after the last LXA_4_ treatment, as expected, LXA_4_ tissue level did not change. We added discussion (Discussion, second paragraph) to clarify that we cannot directly prove that LXA_4_ treatment is increasing LXA_4_ levels in the inguinal lymph nodes or the eye.

**Author response image 2. respfig2:** Quantification of LXA_4_ in EAU-challenged mice treated with LXA_4_ or vehicle control. LXA_4_, 5-HETE, 12-HETE, 15-HETE in pg per sample in inguinal lymph nodes of EAU-challenged mice harvested on day 10 and day 16 post-immunization, quantified by LC-MS/MS. n=4-5 per group.

2) Provide evidence or discussion to show that other lipid mediators (i.e. leukotrienes and prostaglandins) are not affected in the ALOX5-deficient mice.

The Alox5^-/-^ mouse line used in our experiments was generated in 1994 (7) and is an established Jackson Laboratory strain that has been used in numerous publications to study the 5-LOX pathway in disease models. Deficient LTB_4_, peptido-leukotrienes and LXA_4_ formation has been established for this mouse line. We are not aware of any publications that have reported marked compensatory upregulation of other eicosanoid pathways (COX, 12-LOX or 15-LOX) in this mouse line. In response to the reviewer’s concern, we performed lipidomic analysis (new Figure 2—figure supplement 1 and Results subsection “LXA_4_ limits development and progression of EAU”) but did not detect significant differences in basal inguinal lymph node levels of 12-HETE, 15-HETE, PGE2 or PGD2 in Alox5^-/-^ mice. In addition, in WT mice LTB_4_ levels did not significantly change in lymph nodes or the eye in EAU (new Figure 1—figure supplement 1B). Therefore, we focused on LXA_4_, which is selectively down-regulated in inguinal lymph nodes during the development of an adaptive immune response.

3) Examine if LXA4 treatment of isolated T cells in vitro can impact the gene expression levels of chemokine receptors and S1pr1, like they did in the glycolysis analysis (Figure 6).

As requested, we have carried out additional experiments to examine the direct action of LXA_4_ with T cells in vitro. Flow plots shown in the new Figure 4F-G and Results (subsection “LXA_4_ modulates T cell trafficking in inguinal lymph nodes that drain the immunization site”) show that consistent with our hypothesis, LXA_4_ treatment of anti-CD3 and anti-CD28 stimulated T cells leads to increase in CCR7 and decrease in S1PR1 expression.

References:

1) Arnardottir, H. H., Dalli, J., Norling, L. V., Colas, R. A., Perretti, M., and Serhan, C. N. (2016) Resolvin D3 Is Dysregulated in Arthritis and Reduces Arthritic Inflammation. J Immunol 197, 2362-2368

2) Becares, N., Harmala, S., China, L., Colas, R. A., Maini, A. A., Bennet, K., Skene, S. S., Shabir, Z., Dalli, J., and O'Brien, A. (2019) Immune Regulatory Mediators in Plasma from Patients with Acute Decompensation are Associated With 3-month Mortality. Clin Gastroenterol Hepatol

3) Fosshaug, L. E., Colas, R. A., Anstensrud, A. K., Gregersen, I., Nymo, S., Sagen, E. L., Michelsen, A., Vinge, L. E., Oie, E., Gullestad, L., Halvorsen, B., Hansen, T. V., Aukrust, P., Dalli, J., and Yndestad, A. (2019) Early increase of specialized pro-resolving lipid mediators in patients with ST-elevation myocardial infarction. EBioMedicine 46, 264-273

4) Shivakoti, R., Dalli, J., Kadam, D., Gaikwad, S., Barthwal, M., Colas, R. A., Mazzacuva, F., Lokhande, R., Dharmshale, S., Bharadwaj, R., Kagal, A., Pradhan, N., Deshmukh, S., Atre, S., Sahasrabudhe, T., Kakrani, A., Kulkarni, V., Raskar, S., Suryavanshi, N., Chon, S., Gupte, A., Gupta, A., Gupte, N., Arriaga, M. B., Fukutani, K. F., Andrade, B. B., Golub, J. E., and Mave, V. (2019) Lipid mediators of inflammation and Resolution in individuals with tuberculosis and tuberculosis-Diabetes. Prostaglandins and other lipid mediators 147, 106398

5) Gao, Y., Min, K., Zhang, Y., Su, J., Greenwood, M., and Gronert, K. (2015) Female-Specific Downregulation of Tissue Polymorphonuclear Neutrophils Drives Impaired Regulatory T Cell and Amplified Effector T Cell Responses in Autoimmune Dry Eye Disease. J Immunol 195, 3086-3099

6) Gao, Y., Su, J., Zhang, Y., Chan, A., Sin, J. H., Wu, D., Min, K., and Gronert, K. (2018) Dietary DHA amplifies LXA_4_ circuits in tissues and lymph node PMN and is protective in immune-driven dry eye disease. Mucosal immunology 11, 1674-1683

7) Chen, X. S., Sheller, J. R., Johnson, E. N., and Funk, C. D. (1994) Role of leukotrienes revealed by targeted disruption of the 5-lipoxygenase gene. Nature 372, 179-182